# Audio–Visual Fusion Based on Interactive Attention for Person Verification

**DOI:** 10.3390/s23249845

**Published:** 2023-12-15

**Authors:** Xuebin Jing, Liang He, Zhida Song, Shaolei Wang

**Affiliations:** 1School of Computer Science and Technology, Xinjiang University, Urumqi 830017, China; 17693089196@163.com (X.J.); szd@stu.xju.edu.cn (Z.S.); wangsl0407@stu.xju.edu.cn (S.W.); 2Xinjiang Key Laboratory of Signal Detection and Processing, Urumqi 830017, China; 3Department of Electronic Engineering, and Beijing National Research Center for Information Science and Technology, Tsinghua University, Beijing 100084, China

**Keywords:** face verification, speaker verification, audio–visual fusion, attention, gated, inter–attention

## Abstract

With the rapid development of multimedia technology, personnel verification systems have become increasingly important in the security field and identity verification. However, unimodal verification systems have performance bottlenecks in complex scenarios, thus triggering the need for multimodal feature fusion methods. The main problem with audio–visual multimodal feature fusion is how to effectively integrate information from different modalities to improve the accuracy and robustness of the system for individual identity. In this paper, we focus on how to improve multimodal person verification systems and how to combine audio and visual features. In this study, we use pretrained models to extract the embeddings from each modality and then perform fusion model experiments based on these embeddings. The baseline approach in this paper involves taking the fusion feature and passing it through a fully connected (FC) layer. Building upon this baseline, we propose three fusion models based on attentional mechanisms: attention, gated, and inter–attention. These fusion models are trained on the VoxCeleb1 development set and tested on the evaluation sets of the VoxCeleb1, NIST SRE19, and CNC-AV datasets. On the VoxCeleb1 dataset, the best system performance achieved in this study was an equal error rate (EER) of 0.23% and a detection cost function (minDCF) of 0.011. On the evaluation set of NIST SRE19, the EER was 2.60% and the minDCF was 0.283. On the evaluation set of the CNC-AV set, the EER was 11.30% and the minDCF was 0.443. These experimental results strongly demonstrate that the proposed fusion method can significantly improve the performance of multimodal character verification systems.

## 1. Introduction

Person verification technology based on biometrics has been widely used in applications, such as intelligent gates, banking, and forensic investigations. Speaker verification (SV) and face verification (FV) tasks have emerged as hot research topics. These tasks involve studying and testing two typical biometric features: audio and face. With the recent advancements in deep learning, SV and FV have been pushed to the forefront, resulting in significant improvements in their performance. Over the past few years, researchers have proposed different deep neural network architectures, including ResNet34 [1], the Time-Delay Neural Network (TDNN) [2], and the Extended Context Aggregation-Passthrough Time-Delay Neural Network (ECAPA-TDNN) [3]. Additionally, various loss functions, such as marge loss [4], triplet loss [5], and AM-Softmax [6], have been explored. Some systems have demonstrated impressive performance and can even be applied in real-life scenarios.

Despite the significant advancements in the SV and FV systems, their performance can deteriorate sharply under more challenging conditions. In real-world applications, voice-based SV systems often encounter issues, such as channel distortion or noise interference. On the other hand, image-based FV systems face challenges related to illumination variations, facial motion, and changes in posture, among others. Therefore, it is necessary to incorporate audio and visual contexts to obtain a robust person recognition system.

Currently, the state-of-the-art fusion models for multimodal person recognition utilize separate feature extraction networks to embed audio and face modalities into a deep feature space. These features, either combined through simple concatenation with weighted fusion or using score-level fusion, lack robustness against degradation and fail to capture feature quality. Intuitively, an ideal audio–visual fusion recognition system should simultaneously incorporate audio and face modalities, giving more weight to the modality with better discriminative features. The weight allocation should be robust, enabling its applicability to individuals with different accents and facial appearances, accounting for variations in the audio and visual backgrounds, and compensating for missing or corrupted modalities. The comparison of audio–visual fusion technologies is shown in Table 1:

Recently, an attention mechanism based on the features themselves has been used to aggregate the features of each modality according to their quality [7]. However, training the feature extractors aims to extract unique feature vectors specific to that modality, not the feature quality. In the study by Rao et al. [8], they found that aggregating a set of face images into a single image and extracting the features from it outperformed aggregating the features from all the images. Therefore, estimating the feature quality based on higher-level features would lead to better performance.

In this study, we extract the unimodal feature vectors separately using a pretrained model and then output the fused feature vectors using a fusion network. We introduce the attention mechanism in the audio–visual feature fusion process, which effectively fuses the multimodal data and focuses on the most informative region to improve recognition accuracy. The attention mechanisms used in this paper are all self-attentive, meaning they utilize their information to calculate attention weights. We mainly propose four generalized audio–visual multimodal architectures with deep neural networks, in order: Concat Feature Fusion, Attention Feature Fusion, Gated Feature Fusion, and Inter–Attention Feature Fusion. Therefore, the main contributions and novelty of our work are listed as follows:We propose a gated-based multimodal feature fusion model that provides a flexible and effective way to control the flow of information between different modalities in audio–visual fusion. By using this method, we can better fuse speech and facial features, thus improving the recognition system’s performance.We propose a multimodal feature fusion model based on interactive attention. This model fosters a greater interaction between the two modalities than traditional attention mechanisms when calculating attention scores. After processing the unimodal and concatenated features using interactive attention, we add the unimodal feature vectors transformed by a fully connected layer to the feature vectors computed through interactive attention. The advantage of this approach lies in its ability to better preserve the original information, avoiding the loss of important features during the feature fusion process.The proposed models have the advantage of being easy to implement and optimize quickly, as all operations are performed in the feature vector space. For example, in this study, we utilize preprepared feature vectors to train the fusion network, which greatly reduces the experimental time.The fusion models we propose are decoupled from the front-end extractors, allowing them to be generalized to various feature vectors extracted by pretrained models. Because they are decoupled from the front end, our method can be applied to different modalities and pretrained models. For instance, when extracting feature vectors from other modalities, we only need to ensure consistency in the data format with the subsequent stage, allowing us to train other fusion modality experiments using this fusion model or to extract feature vectors we require using better pretrained models, further improving our experimental performance.

The remainder of this paper is as follows: The second section describes the related work on single-mode verification tasks and multi-mode fusion verification. The third section focuses on the fusion system, explaining its design and implementation. The fourth section presents the experimental results and analysis conducted. The fifth and final section discusses the conclusion drawn from the study and the prospect of future work.

## 2. Related Work

### 2.1. Face Verification

Put simply, face verification involves using a feature extractor to extract image information and comparing it with pre-registered image data. When the comparison exceeds a certain threshold, the images belong to the same person; otherwise, they are considered different individuals. In recent years, the development of FV has seen significant advancements. Notably, AlexNet [9], designed by Geoffrey Hinton, the winner of the 2012 ImageNet competition, and his student Alex Krizhevsky, achieved remarkable success. Methods based on deep convolutional neural networks (DCNNs) have become the mainstream for FV tasks. For instance, Facebook’s DeepFace [10] in 2016 demonstrated the highest performance, as evaluated on the challenging LFW dataset [11]. Furthermore, researchers have explored various DCNN-based architectures for FV, resulting in substantial improvements. Notable examples include the DeepID series [12,13], VGGFace [14], FaceNet [15], and Google’s 2018 arcface [4]. Our work, uses FaceNet [15] as the feature extractor to obtain the feature vectors from the images.

### 2.2. Speaker Verification

Similar to face verification discussed earlier, speaker verification involves inputting a voice, extracting voice information, and comparing it with previously registered voice data. If the comparison exceeds a certain threshold, it is determined that the input and registered voices belong to the same person; otherwise, they are considered different individuals. With the advancement of deep neural networks (DNNs), speaker verification modeling has transitioned from traditional approaches like the Gaussian mixture model–universal background model (GMM-UBM) [16] and i-vector [17] to deep speaker-embedding representation learning. A commonly used method for speaker-embedding learning is the d-vector [18], where a fully connected DNN is employed to extract depth features at the frame level, which are then averaged to obtain a speaker representation at the utterance level. Another popular method for speaker-embedding learning is the x-vector [19], based on the time-delay deep neural network (TDNN), which has demonstrated state-of-the-art performance [1] on various datasets. More recently, advanced neural architectures like ResNet and ECAPA-TDNN have further improved speaker verification performance. Our work, uses the ECAPA-TDNN model as a feature extractor to obtain speaker-embedding vectors from audio streams.

### 2.3. Audio–Visual Person Verification

Although SV and FV have made significant progress in recent years, their performance has significantly declined under more challenging conditions. For instance, SV is not robust against background noise, changes in acoustic characteristics caused by human emotion, recording device distance, and other factors. Similarly, FV is also susceptible to factors, such as illumination, posture, emotion, and distance. Due to the varying degrees of disturbances in these two verification methods, researchers have turned to the fusion of the two modalities. The initial work in this direction adopts the strategy of feature-level fusion [20,21,22], where the scores obtained from separately trained unimodal models are combined. Recent research has explored attention-based fusion techniques [7,23,24,25,26], which intelligently combine salient features from input patterns. In general, multimodal systems [27] have been more accurate and robust than unimodal verification systems, especially in noisy conditions. Several modal fusion techniques have been well developed in various domains in recent years. For example, multimodal emotion recognition [28,29,30], multimodal interaction [31], and lip-sync fusion [32,33,34,35] have seen significant advancements. In 2023, Qin et al. [36] provided a comprehensive review of identification techniques and applications in unimodal identity recognition. The review starts by discussing the concepts and limitations of unimodal identity recognition and the motivation and advantages of multi-channel identity recognition. The authors then examine identification technology and applications from four key aspects: the feature level, matching level, decision level, and hierarchical level. Additionally, they discuss security issues and outline future research directions in learning-based identity recognition. In 2013, the framework proposed by John and Kawanishi [37] enhanced the robustness of audio–visual individual recognition by utilizing audio-based individual attributes and a multi-head attention transformer-based network, termed the CNN Transformer Network (CTNet), which achieves excellent performance even in the absence of visual modalities. To facilitate research in audio–visual verification, several multimedia datasets have been introduced. These include the VAST dataset [38], JANUS dataset [20], VoxCeleb1 dataset [39], VoxCeleb2 dataset [40], SpeakingFaces dataset [41], and CN-Celeb-AV dataset [42]. These datasets provide valuable resources for training and evaluating audio–visual verification systems. Furthermore, the 2019 NIST Speaker Identification Assessment (SRE) Challenge [43] has introduced multimodal racing tracks and organized speaker verification challenges. This initiative aims to provide a platform for multimodal person verification research to learn from each other and foster advancements in the field. Based on the above methods and datasets of audio–visual fusion models, this paper proposes to introduce two methods, gated and inter–attention, in audio–visual fusion models. The proposed model shows the advantages in personnel verification and generates high-quality fusion features under the respective characteristics in the case of one modality missing or poor quality, and the generalization of the model is discussed through the comparison of different datasets.

## 3. System

This section will describe the proposed multimodal fusion approach and its audio and visual representation subsystems. Our method utilizes a feature-level fusion approach based on neural network models. Given the discriminative face and speaker representations extracted from each subsystem, our attention layer evaluates the contributions of each representation. We combine them according to the estimated contributions, resulting in a joint representation. During the testing phase, we compute the similarity of the joint representations between the enrollment and test the samples to verify the identities. The overall network structure of this paper is illustrated in Figure 1.

### 3.1. Feature Extractor

#### 3.1.1. Face Feature Extractor

FaceNet [15], presented by Google at CVPR in 2015, proposes a unified framework for solving problems, such as face recognition, verification, and clustering. In this framework, all these tasks can be treated as feature-based problems, where the main focus is effectively mapping faces to a feature space. The fundamental idea is to learn a mapping from face images to a 512-dimensional Euclidean space using convolutional neural networks (CNNs) [44]. This mapping enables face images to be represented as 512-dimensional feature vectors.

Inspired by the concept of a correlation coefficient in two-dimensional space, the similarity between face images is characterized by the inverse distance between their feature vectors. For instance, feature vectors of different images of the same individual have smaller distances, while feature vectors of images belonging to different individuals have larger distances. By leveraging the similarity measures between the feature vectors, FaceNet addresses the problems of face recognition, verification, and clustering.

#### 3.1.2. Speaker Feature Extractor

ECAPA-TDNN was proposed by Desplanques et al. [3] at the University of Gothic, Belgium, in 2020. The scheme achieved first place in the International Voice Recognition Competition by introducing the squeeze–excitation (SE) module and the channel attention mechanism (VoxSRC2020). The ECAPA-TDNN model consists of the following main modules:SE-Res2Block: Res2Net Block + SE block;Multi-layer feature aggregation and summation;Attentive statistic pooling.

#### 3.1.3. Extractor Parameters

To obtain audio ($e_{a}$) and visual ($e_{v}$) embeddings, we extracted the audio and visual embeddings for training and testing using the unimodal system described in Section 3.1.

For face feature extraction, we utilize the FaceNet model [15], which was trained on the CASIA-WebFace [45] datasets. We used the multitask convolutional neural network (MTCNN) face detector to align and crop the faces. We set certain parameters to ensure consistent results when performing alignment and cropping. We set the input image size to 160×160 pixels, as the FaceNet model requires for fixed-size inputs. Additionally, we set the margin to 0, indicating no extra padding around the detected face. This margin parameter can be used to expand or shrink the facial region for better alignment or subsequent processing. To control the MTCNN face detector, we also set the minimum face size to 20×20 pixels. This parameter defines the smallest face size that can be detected. Any face region smaller than this size will not be detected. Threshold parameters control the thresholds for different stages of detection in the MTCNN. Here, we set the thresholds to 0.6, 0.7, and 0.7, affecting the results of the face detection and keypoint detection. Adjusting these thresholds allows us to balance the trade-off between accuracy and recall. The scaling factor is a parameter used in creating a multi-scale image pyramid. We set the scaling factor to 0.709, which is the recommended default value for MTCNN. The post_process parameter controls whether post-processing operations are performed. Post-processing is used to further process the results obtained from the face and keypoint detectors, such as face alignment or pose estimation. Here, we set post_process to True, indicating that post-processing operations are performed to obtain more accurate face alignment results. Then, we input the aligned and cropped face frames into the FaceNet model to obtain a 512-dimensional embedding vector for each face. These embedding vectors can calculate the similarity between faces or for tasks, such as face recognition.

For speaker feature extraction, we utilize the ECAPA-TDNN model [3]. The procedure is as follows: we randomly extract a 2-s audio segment from the original audio. If the extracted segment is shorter than 2 s, we repeat the preceding audio to make it reach the 2-s mark. The input features comprise frames with a length of 25ms and a frame shift of 10ms, resulting in 80-dimensional Fbanks. We apply SpecAugment feature enhancement techniques. Subsequently, the audio features pass through the standard ECAPA-TDNN, which includes three SE-Res2Block modules. The channel size is set to 1024. The features then undergo attentive statistics pooling (ASP) through a pooling layer, followed by a fully connected layer to yield embeddings of a dimensionality of 192. We employ the AAM-Softmax loss function [4], with a margin of 0.2 and a scale of 30.

### 3.2. Fusion Method

For the fusion module, the model inputs the speaker and face embeddings obtained from the SV and FV pretraining model extractors. The embedded fusion network is constructed and optimized using these speaker and face embeddings to generate the output of the two-mode fusion embedding. Before concatenating the embeddings, it is necessary to L2-normalize both embeddings obtained from the pretraining models. This step ensures that the transformed embeddings reside in a co-embedded space, which is more suitable for subsequent combinations. The models presented in this paper are free for researchers to use and can be downloaded from https://github.com/jingxuebin20/AVF#avf, accessed on 10 October 2023.

#### 3.2.1. Concat Feature Fusion

The Concat Feature Fusion model is a simple fusion method that utilizes a unimodal system model to extract and process feature vectors. The feature vectors from the two modalities are concatenated to form a feature vector with a dimension of 704 (192 + 512). To train the model similarly to other fused features, a fully connected layer with 512 hidden nodes is used, followed by a dropout layer to prevent overfitting, and a ReLU activation function is applied. The model’s structure is depicted in Figure 2a.

The advantage of this method lies in its simplicity and suitability for achieving rapid modality fusion, leading to good results in certain scenarios. It is commonly used as a baseline benchmark for fusion models. In this study, it is also employed as a baseline for a performance comparison with several other fusion methods. However, the limitation of this approach is that, although it employs a fully connected layer to learn the relationship between the two modalities after feature concatenation, it still exhibits a relatively weak interaction between the two modalities.

#### 3.2.2. Attention Feature Fusion

We have developed a multimodal attention model to focus on the salient modalities in the input while generating a robust fusion representation suitable for person verification tasks. This model draws inspiration from the multisensory capabilities of humans. In the human multisensory system, selective attention exists [46] to enable individuals to prioritize and select key information from complex sensory inputs. The human attentional mechanism dynamically extracts salient features as needed, without collapsing the overall information into vague generalizations. This process is known as Attention Feature Fusion (AFF). Our model is depicted in Figure 2b.

Implementations of this attention mechanism in deep neural networks have proven successful in various machine learning applications. The attention network we utilize is similar to a differentiable soft attention network. While previous studies have predominantly focused on spatial or temporal attention, we extend the attention mechanism to concentrate on the relationship between different modalities. Given the face and speaker feature vectors $e_{a}$ and $e_{v}$ extracted from the pretrained system, we define the relationship between different modalities using the attention layer $f_{\mathrm{att}}\left(.\right)$ to determine the attention fraction ${\hat{a}}_{\left\{a,v\right\}}$:(1)$${\hat{a}}_{\left\{a,v\right\}}=f_{\mathrm{att}\;}\left(\left[e_{a},e_{v}\right]\right)=W^{⊤}\left[e_{a},e_{v}\right]+b$$
where $W\in R^{m\times d}$ and $b\in R^{m}$ are the learnable parameters of the attention layer, *m* and d represent the number of fusion modes and the input dimensions of the attention layer, respectively, and $e_{a}$ and $e_{v}$ are the embedding of the speaker and the face. Then, the fusion ${\mathrm{Embedding}}_{out}$ embedding is obtained from the weighted sum: (2)$${\mathrm{Embedding}}_{out}=\sum_{i\in \{a,v\}}\alpha_{i}{\tilde{e}}_{i},\;where\;\alpha_{i}=\frac{\exp\left({\hat{a}}_{i}\right)}{\sum_{k\in \{a,v\}}\exp\left({\hat{a}}_{k}\right)},\;i\in \left\{a,v\right\}$$
where $\tilde{e}$ indicates that the projection is embedded into a co-embedded space compatible with the linear combination. To map ${\tilde{e}}_{\left\{a,v\right\}}$ from $e_{\left\{a,v\right\}}$, we used an FC layer with 512 hidden nodes, that is, $\tilde{e}\in R^{512}$. We do not use nonlinearity in the FC layer. We train the attention network by jointly embedding ${\mathrm{Embedding}}_{out}\in R^{512}$ using loss functions.

Introducing the attention network enables us to handle damaged or missing data from any modality more naturally. In traditional multimodal tasks, it is usually necessary to perform integrity and quality checks and the corresponding preprocessing for each modality’s data. However, in our framework, the attention network can automatically assess the quality and credibility of the given multimodal data without explicit specification. For example, when the audio signal is heavily disturbed by surrounding noise, the attention network automatically deactivates the path for audio representation based on learned attention weights and relies solely on the facial representation for the task. Similarly, if the facial image has significant occlusions or a low image quality, the attention network adjusts accordingly, primarily relying on audio representation. This adaptive attention mechanism enables the model to adjust flexibly based on the input data’s credibility and informational value, enhancing the system robustness and performance in complex environments.

By leveraging the attention network, our model can selectively and dependently utilize effective inputs within multimodal data. The attention network can discover inherent correlations and complementarity among different modalities, focusing attention on the most beneficial modalities for the task. This attention mechanism makes our model more robust and adaptable, effectively handling damaged, missing, or other exceptional data conditions for person verification tasks in multimodal data.

Therefore, by incorporating the attention network, our model can automatically evaluate the quality of multimodal data and allocate adaptive attention based on the credibility and informational value of the data. This attention mechanism enables the model to handle damaged or missing data from different modalities more naturally and to selectively rely on effective inputs within multimodal data. This enhances the robustness and performance of person verification tasks.

#### 3.2.3. Gated Feature Fusion

In this section, we present a method called Gated Feature Fusion (GFF) that employs a multiplicative gate to control the flow of information in the audio and visual modalities. This idea is inspired by the flow control mechanism found in recurrent neural networks, like GRU or LSTM. A similar concept has been applied to fuse information from image and text modalities [47].

The paper above noted that the gated multimodal unit possesses an interesting property. As a microscopic operation, it can easily integrate with other neural network structures and can be trained using standard gradient-based optimization algorithms. In this paper, we extend this idea to audio–visual fusion. Figure 2c depicts the gated multimodal fusion architecture. In this architecture, we employ the feature extractor introduced in Section 3.1 to extract the feature vectors $e_{a}$ and $e_{v}$ from a given face and voice, respectively. The learnable parameter $W_{z}$ is used to compute the gate vector $z\in R^{D}$:(3)$$z=\sigma \left(W_{z}\left(\left[e_{a},e_{v}\right]\right)\right)$$

Then, we use the gate vector z to merge ${\tilde{e}}_{a}$ and ${\tilde{e}}_{v}$ into ${\mathrm{Embedding}}_{out}$, where ⊙ represents the element-by-element product: (4)$${\mathrm{Embedding}}_{out}=z⊙\tanh\left({\tilde{e}}_{a}\right)+\left(1-z\right)⊙\tanh\left({\tilde{e}}_{v}\right)$$

The gated multimodal feature fusion approach offers a flexible and effective method to fuse audio and visual information while controlling the flow of information between different modalities. With this approach, we can achieve improved fusion of audio and visual features, leading to enhanced performance in recognition systems.

#### 3.2.4. Inter–Attention Feature Fusion

In this section, we apply the inter–attention mechanism to audio–visual fusion to efficiently extract information from $e_{a}$ and $e_{v}$ and computing the inter–attention score. We refer to this network as Inter–Attention Feature Fusion (IAFF). Inter–attention mechanisms are widely used in tasks, such as multimodal emotion recognition. This is because multimodal emotion recognition involves extracting a large number of features. However, an increase in the number of features not only leads to an increase in training parameters and noise generation but can also result in the loss of critical information.

Similarly, in audio–visual fusion, there may be a problem of information loss in a certain modality or poor quality of features. Therefore, it becomes necessary to focus on the most important features. Introducing an interactive attention mechanism is an effective approach to address this issue. The mechanism used in this paper is the self-attention mechanism, which utilizes information from within the modality itself to calculate attention weights. Traditional attention mechanisms usually rely on external information for calculating attention weights. We can generate a more representative vector of the person’s characteristics by obtaining more effective information from each modality. The structure of the proposed interactive attentional feature fusion network is shown in Figure 2d. This method offers the following advantages:1.The attention scores of interactive attention, which are calculated interactively in both modalities, are more interactive than the simple attention mechanism;2.After processing the unimodal and multimodal information, we again add the unimodal information to the multimodal information through the FC. This operation has the advantage of preventing the loss of critical information.

This converged network is calculated as follows:(5)$$U_{a}=DP\left(\tilde{e_{a}}+softmax\left(\frac{{\tilde{e_{a}}}^{⊤}a_{a}}{\sqrt{d_{a}}}\right)a_{a}\right)$$
(6)$$U_{v}=DP\left(\tilde{e_{v}}+softmax\left(\frac{{\tilde{e_{v}}}^{⊤}a_{a}}{\sqrt{d_{a}}}\right)a_{v}\right)$$
(7)$${\mathrm{Embedding}}_{out}=DP\left(U_{a}+U_{v}\right)$$

## 4. Experimental Setup

This section describes the details of the lab settings for the system. We use cosine similarity in the scoring stages of all the experiments in this work, and the evaluation indicators are the equal error rate (EER) and minimum detection cost function (minDCF).

### 4.1. Dataset

In our experiment, we utilized audio and visual data from three datasets, which are all publicly available standard audio–visual datasets, namely, VoxCeleb1 [39], the NIST SRE19 multimedia dataset [48], and CN-Celeb-AV (CNC-AV) [42]. We used the development part of the VoxCeleb1 dataset for the training set, which comprises 1211 speakers and 148,642 utterances. In the test set, we employed the test set of VoxCeleb1, the NIST SRE19 multimedia dataset, and the CNC-AV-Eval-F dataset.

The NIST SRE19 dataset provides two multimedia datasets for the audio and visual tracks: the JANUS Multimedia Dataset (LDC2019E55) and the 2019 NIST Speaker Recognition Evaluation Audio-Visual Development Set (LDC2019E56). We specifically used the LDC2019E55 dataset for this experiment, which served as the evaluation set. LDC2019E55 is a JANUS Multimedia Dataset extracted from the IARPA JANUS Benchmark-B (IJB-B) dataset [48], as described in detail in the article [20]. The purpose of the CNC-AV dataset is to evaluate the real performance of audio–visual speaker recognition (AVPR) technology under unconstrained conditions and to provide a standard benchmark for AVPR research. All the data are collected from Bilibili (https://www.bilibili.com/, accessed on 1 September 2023), a popular Chinese public medium. Overall, it contains more than 419k video clips (669 h) from 1136 people (mainly Chinese celebrities) covering 11 types in CN-Celeb [42]. CNC-AV-Eval-F is a subset of the CNC-AV dataset, which contains 197 speakers. Most of the data in this dataset contain both audio and visual information. It is a standard evaluation set for ‘full-modality’ person recognition systems. The statistical tables of the datasets can be found in Table 2, Table 3 and Table 4.

#### Experimental Parameters

The audio and visual embeddings, mentioned previously, are used in the training process. The combined spliced embedding has a dimension of 704 (192 + 512). All the embeddings undergo L2 normalization as a preprocessing step. Regarding the attention-based fusion systems described in Section 3.2, the transformation layer consists of an FC layer with 512 units, and the output embedding dimension is set to 512.

During training, the batch size is 64, and the learning rate is 0.0001. The experiments are conducted using the Adam [49] and SGD optimizers separately to compare their differences in model convergence speed and performance. Adam performs better than SGD and does not require manual tuning of the learning rate. By using four loss functions, Cross-entropy [50], Center loss [51], AM-Softmax [6], and AAM-Softmax [4], and evaluating their performance, AM-Softmax was the best performer in the experiment, with a margin of 0.2 and a scale of 30. Loss functions calculate the distance between the embeddings (fused feature vectors) and the label. The final output is obtained by fusing the embeddings and selecting the output of the fully connected layer. In constructing the training pairs, the speaker and face embeddings are randomly selected from the same individual. All the systems undergo 60 epochs of training. The training is performed on one NVIDIA GeForce RTX 3090 GPU.

### 4.2. Experimental Results and Analysis

The section provides a comprehensive evaluation, comparison, and analysis of the newly proposed approaches for person verification.

#### 4.2.1. Analysis of Unimodal Experimental Results

The experimental results of the audio–visual unimodal speaker verification system, using the AM-Softmax loss function, are presented in Table 5. The table presents the results obtained using the ECAPA-TDNN audio feature extractor and the FaceNet and Resnet50 visual feature extractors. The evaluation sets include VoxCeleb1 and the NIST SRE19 and CNC-AV multimedia datasets.

Based on the data in Table 5, it is evident that in the domain of face verification models, the FaceNet model outperforms the Resnet50 model. This superiority can be attributed to several key factors. The outstanding performance of FaceNet in face verification tasks primarily stems from its purpose-built architecture, the utilization of the triplet loss function, the distance measurement in the embedding space, and the implementation of more rigorous training strategies. These elements collectively establish FaceNet as a robust facial verification model. Compared to general-purpose deep learning models such as Resnet50, FaceNet is better suited for tasks related to facial analysis.

In the analysis of the three datasets, we can observe that our model performs best on the VoxCeleb1 dataset, followed by the NIST SRE19 multimedia dataset, and exhibits the lowest performance on the CNC-AV dataset. This outcome may be attributed to the following reasons:Language Discrepancy in Datasets: Our model was trained using the development set from the VoxCeleb1 dataset, primarily comprising English audio data. In contrast, the CNC-AV dataset mainly consists of Chinese audio data. This linguistic distinction could lead to performance variations, as language features play a significant role in speech recognition.Limitations in Model Generalization: Another potential factor is that the chosen model may exhibit limited generalization across different datasets. If the model’s generalization performance across various languages and data types is subpar, it may underperform on specific datasets, especially when pronounced differences exist between them.

#### 4.2.2. Analysis of Experimental Results of Audio–Visual Fusion

The experimental results of the audio–visual multimodal fusion person verification system under several models are shown in Table 6.

##### Performance Analysis under Different Models

Compared to single-modal approaches, the experimental results from the testing phase on the three datasets demonstrate that utilizing fusion models can significantly enhance verification performance. Our experiments indicate that even employing the simplest fusion method, such as CFF, we can achieve better results than single-channel systems. This implies that the fusion of audio and visual cues can effectively improve identity verification performance.

On the VoxCeleb1 dataset, it can be observed that GFF performs the best in terms of performance, it is shown in bold in the Table 6, followed by IAFF, AFF, and CFF. Theoretically, this result is reasonable because fusion methods based on AFF are theoretically superior to those based on CFF. The attention mechanism can generate more expressive fusion feature vectors, focusing on crucial channels from multiple inputs. By prioritizing important information in the data, the attention mechanism can better capture key details, thereby enhancing system performance. GFF outperforms AFF because this gating multimodal feature fusion method provides a flexible and effective way for audio–visual fusion, controlling the flow of information between different modalities. By employing this method, we can better integrate speech and facial features, thus improving the recognition system’s performance. IAFF outperforms the basic AFF because there is a stronger interaction between these two modalities when computing the attention scores for interactive attention. Compared to traditional attention mechanisms, this can more accurately capture the correlated information between modalities.

We observe that IAFF exhibits the best performance on the NIST dataset, it is shown in bold in the Table 6, followed by GFF, AFF, and CFF. This result is theoretically sound, and its reasoning is similar to the analysis in the VoxCeleb1 dataset.

Observing the experimental results of the CNC-AV dataset, we noticed that the performance of our proposed fusion model on this dataset was not significant. It even lagged behind the baseline model slightly. This indicates that our model might be sensitive to different languages, and its generalization ability is relatively poor, especially on Chinese datasets. Through this round of experiments, we have once again confirmed the limitation of our model in terms of generalization performance. Therefore, our next step will focus on training a fusion model with a stronger generalization ability.

This research stems from the initial idea of a lab project to explore the application of audio–visual fusion methods for person verification and speech classification in the short video domain. The background of this project is to solve the problem of person verification and speech classification on short video platforms to improve the user experience and platform functionality. By combining audio and visual fusion techniques, we are promoting more efficient, accurate, and real-time people verification in the short video domain.

In our research, we are actively involved and integrated in this lab project. By employing the audio–visual fusion model proposed in this paper, we explored the prospects for a wide range of applications of this approach in the short video domain. The project aims to successfully integrate the research results into short video platforms to provide a more realistic, interesting, and safe user experience while addressing the challenges of real-world applications.

##### Performance Comparisons with Other Algorithms

In our study, we compared systems that used different loss functions and fusion models and compared the results with previous studies. Table 6 presents the specific comparison results, indicating that our fusion model outperformed the previous study on the VoxCeleb1 test set.

This difference can be attributed to several factors, including variations in the training set and differences in the experimental setup. As mentioned earlier, we used a different training set, and our experimental setup may have differed from the previous study. These factors might have contributed to the superior performance of our fusion model.

Furthermore, our fusion model may have more accurately captured critical features when dealing with multimodal data. The design of the fusion model might have better modeled the relationships between different modalities, resulting in improved performance. For example, we may have employed a more powerful network architecture or more effective training strategies to handle audio and visual data fusion.

However, under the CNC-AV dataset, by comparing with the score fusion method in paper [42], it can be seen that the method proposed in this paper does not perform as well as the other methods. As mentioned before, the generalization performance of the model used in this paper is not very good under this dataset. Secondly, the unimodal feature extractor used in [42] is not the same as in this paper.

Overall, our research findings indicate that our fusion model exhibits better performance on the VoxCeleb1 test set within systems utilizing different loss functions and fusion models. This advantage could be attributed to a combination of factors, including the choice of training set, optimization of the experimental setup, and the superiority of the fusion model in handling multimodal data. However, the results are not satisfactory under the CNC-AV dataset, reflecting that the generalization performance of the model in this paper needs to be improved.

## 5. Conclusions

In this work, we have developed various architectures and strategies to explore audio–visual multimodal fusion for person validation. Building upon the baseline system, we designed three fusion networks based on attention: Attention Feature Fusion, Gated Feature Fusion, and Inter–Attention Feature Fusion. The attention mechanisms used in this paper are all self-attentive, meaning they utilize their information to calculate attention weights. This is a departure from traditional attention mechanisms, which often rely on external information for calculating attention weights.

Through experimental verification, we found that these fusion architectures effectively combine the features from the two modalities and significantly enhance the system performance for person verification. While interactive attention is commonly employed in multimodal emotion recognition tasks, this paper represents the first application of interactive attention to the multimodal audio–visual fusion verification task.

We can draw the following conclusions based on the analysis and comparison of the experimental results. All four fusion models proposed in this study significantly improve the performance of multimodal person verification tasks. Our fusion models are decoupled from the front-end feature extractors, allowing for the extraction of modality-specific feature vectors using various pretrained models. Therefore, our fusion models can be generalized to other modalities.

There are two limitations to this study. Firstly, our fusion system requires the pre-extraction of feature vectors for each modality. While pre-extracting features significantly reduces the training time of the fusion model, and the decoupled fusion model allows the use of various pretrained models, in our daily lives, end-to-end models are needed to facilitate the interaction between the two modalities during the front-end feature extraction. Secondly, our model is not well suited for Chinese datasets, and its generalization performance needs improvement.

Future work will develop an end-to-end audio–visual fusion model with better generalization performance and high quality fusion features. First, to improve the generalization performance of the model, we may need more training data across languages and data types and perform data cleaning and preprocessing and data enhancement operations in the data processing and preparation phase. Second, we consider adapting the model architecture to adapt to the diversity of different languages and data types, and third, we adopt early stopping, integrated learning, or adversarial training in the model training strategy to improve the generalization performance and robustness of the model. Finally, using multiple datasets and cross-validation in model evaluation and validation, through these methods, it is believed that an end-to-end model with better generalization performance will be developed. In addition, to generate high-quality fusion features, we plan to use an approach based on the joint training of an interactive attention mechanism and self-encoder network for enhanced audio and visual feature fusion. After interactive attention module feature fusion, a fusion feature loss is generated; this fusion feature is reconstructed into the original speech and face features by the self-encoder, where the process generates a reconstructed audio feature loss and a reconstructed face feature loss. It is suggested to use a multilingual dataset for the joint training of these three loss functions. The system can learn more representative and informative audio and visual fusion features while retaining the original information, which ultimately improves the performance and robustness of the overall personnel verification system.

## Figures and Tables

**Figure 1 sensors-23-09845-f001:**
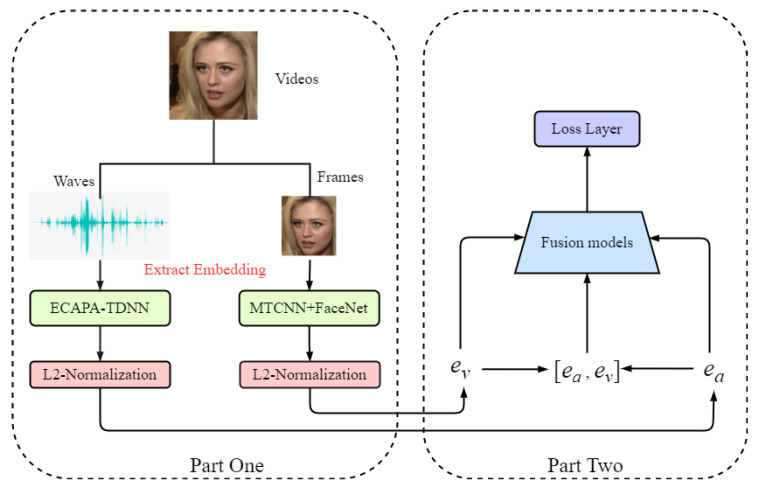
The overall network structure of this paper consists of two parts. The diagram depicts the left part, which extracts the embeddings using the pretraining model. The right part represents the fusion model training phase, where the left part’s output serves as the right part’s input.

**Figure 2 sensors-23-09845-f002:**
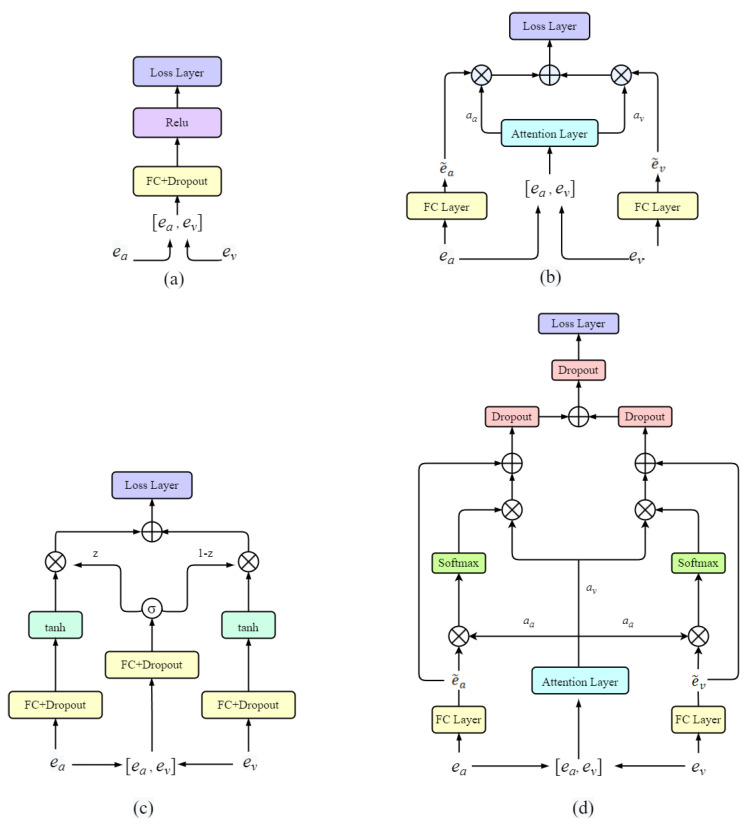
Framework diagram of the fusion model. (**a**) Concat Feature Fusion, (**b**) Attention Feature Fusion, (**c**) Gated Feature Fusion, and (**d**) Inter–Attention Feature Fusion.

**Table 1 sensors-23-09845-t001:** Comparison table of audio–visual fusion methods.

Methods	Advantages	Disadvantages
Early Fusion	(1) Simple, intuitive, and easy to implement. (2) Considers global information for audio and visual.	(1) Localized information for each modality is ignored. (2) Needs to deal with size and representation inconsistencies between different modal data.
Late Fusion	(1) Capable of capturing localized information for each modality. (2) Different network structures can be used to handle different modes.	(1) Cannot fully utilize the correlation between modes. (2) May require more parameters and computational resources.
Mid-level Fusion	(1) Combines the advantages of early fusion and late fusion, taking into account both global and local information. (2) Different levels of fusion can be flexibly selected.	Higher requirements for the design and adaptation of network structures.
Attention-based Fusion	Capable of adaptively capturing key information for each modality.	(1) The training and inference process may be more complex. (2) Additional computational resources are required.

**Table 2 sensors-23-09845-t002:** Statistics for the VoxCeleb1 dataset.

	Dev	Test
# of speakers	1211	40
# of videos	21,819	677
# of utterances	128,642	4708
# of images	1,167,721	39,085

**Table 3 sensors-23-09845-t003:** Statistics for the NIST SRE19 multimedia dataset.

Condition	Split	#Enroll Videos	#Test Videos	#Target	#Nontarget
SRE19	DEV	102	319	244	32,294
	EVAL	258	914	681	235,131

**Table 4 sensors-23-09845-t004:** Statistics for the CNC-AV dataset.

	CNC-AV-Dev-F	CNC-AV-Eval-F	CNC-AV-Eval-P
# of Genres	11	11	11
# of Persons	689	197	250
# of Segments	93,973	17,717	307,973
# of Hours	199,70	41,96	427,74

**Table 5 sensors-23-09845-t005:** Performance comparison of unimodal systems on different test datasets.

Datasets	Modality	System	EER (%)	minDCF
VoxCeleb1	Audio	ECAPA-TDNN	0.98	0.068
	Visual	FaceNet	3.96	0.263
		Resnet50	5.26	0.276
NIST SRE19	Audio	ECAPA-TDNN	7.93	0.484
	Visual	FaceNet	9.28	0.25
		Resnet50	13.85	0.358
CNC-AV	Audio	ECAPA-TDNN	17.04	0.764
	Visual	FaceNet	27.49	0.743
		Resnet50	29.89	0.776

**Table 6 sensors-23-09845-t006:** Performance comparison using different models and datasets.

Dataset	System	EER(%)	minDCF
VoxCeleb1	CFF	0.32	0.023
	AFF	0.30	0.020
	AFF [24]	0.718	-
	GFF	**0.23**	**0.011**
	GFF [24]	0.744	-
	Multi-scale attention [23]	0.64	0.076
	IAFF	0.25	0.018
NIST SRE19	CFF	3.10	0.233
	AFF	2.91	0.286
	GFF	2.64	0.245
	IAFF	**2.60**	0.283
CNC-AV	CFF	**11.30**	**0.443**
	AFF	13.06	0.566
	GFF	12.18	0.532
	IAFF	11.72	0.541
	Score Fusion [42]	8.64	0.271

## Data Availability

The datasets supporting the conclusions of this article can be found at https://www.robots.ox.ac.uk/~vgg/data/voxceleb/vox1.html (accessed on 1 September 2021), https://sre.nist.gov (accessed on 1 September 2021), and http://cnceleb.org (accessed on 1 September 2023).

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
