# Peer review of "Audio–Visual Fusion Based on Interactive Attention for Person Verification"

_sensors, 2023, doi:10.3390/s23249845_

Round 1
Reviewer 1 Report (New Reviewer)
Comments and Suggestions for Authors
The article presents a comprehensive approach to person verification using biometric features like voice and face. It emphasizes the necessity of incorporating both audio and visual contexts to enhance the robustness of person recognition systems, especially under challenging conditions like noise interference or facial motion​. The presented work is comprehensive, and suitable experimental work has been done, but some concerns must be addressed.
· Include a more detailed comparative analysis with existing methods. This will help highlight the proposed system's improvements and advantages over current approaches.
· Elaborate on potential future improvements or extensions of the work.
· If possible, include real-world applications or case studies where this system could be effectively implemented. This would demonstrate the practical utility of the research.
· Some more recent related works can be added. For example:
o Praveen, R. Gnana, et al. "A joint cross-attention model for audio-visual fusion in dimensional emotion recognition." Proceedings of the IEEE/CVF conference on computer vision and pattern recognition. 2022.
o John, Vijay, and Yasutomo Kawanishi. "Audio-Visual Sensor Fusion Framework using Person Attributes Robust to Missing Visual Modality for Person Recognition." International Conference on Multimedia Modeling. Cham: Springer Nature Switzerland, 2023.
o Song, Haozhen, et al. "Multimodal Person Verification based on audio-visual fusion." 2022 10th International Conference on Information Systems and Computing Technology (ISCTech). IEEE, 2022.
Author Response
Dear Reviewer,
Thank you for your valuable suggestions on our articles. We have revised the three suggestions you made and explained them in detail in the following paragraphs.
Comparative analysis with existing methods: We have expanded the comparative analysis with existing methods to emphasize the improvements and advantages of our proposed system over current methods. In lines 475-499, we have added some new paragraphs detailing the differences between our approach and the current approach.
Detailed elaboration of future work: In lines 526-545, we have added more content that delves into the future direction of our research and potential room for improvement. This includes the integration of new technologies, and possible avenues for performance optimization to give the reader a better understanding of the long-term goals and potential impact of our work.
Presentation of practical application cases: In lines 461-472, we add more information about real-world use cases that illustrate potential real-world applications of our system.
Updates to recent related work: We have updated and added to our recent related work in lines 139-141 and 147-151 to ensure that our literature review is up-to-date and comprehensive.
Thank you again for your valuable suggestions, and we believe that these changes can further enhance the quality of the article and the reading experience. Please feel free to let us know if you have any other suggestions or need further revisions.
Sincerely,
Xuebing Jing
Reviewer 2 Report (New Reviewer)
Comments and Suggestions for Authors
First of all, congratulations to the authors for their work.
Indicatively we can judge the work to be good, well structured and well explained. It is possible to completely replicate the work because it is correctly explained and provided.
However, it does require a few adjustments.
Avoid abbreviations not explained in the abstract, such as EER. They make it very difficult for non-experts to read.
I recommend a re-reading because there are some badly placed or poorly understood sentences.
The big job I recommend is to retrain the model with an optimiser on hyper parameters and cost functions. It is not possible to know regardless that the optimiser chosen (adam), as well as the batch size, and the rest of the hyper parameters, are actually the optimal ones for the proposed case.
Also explain extensively the reason for the choices of regulators, softmax, cost functions, etc.
Author Response
Dear Reviewer,
Thank you very much for your comments and valuable suggestions on our paper. We are very pleased that you found our work to be well structured, clearly explained, and capable of being reproduced in its entirety. In terms of your suggestions, we have made changes accordingly to enhance the quality and readability of the paper.
Avoidance of abbreviations not explained in the abstract: We have made changes to ensure that unexplained abbreviations are avoided in the abstract to make it easier for non-expert readers to understand. Make changes in line 14 of the revised version of the paper.
Fixing sentence placement and lack of clarity of understanding: We have carried out a re-reading of the full paper, rearranging and clarifying a number of sentences to ensure that the overall narrative flows better and is easier to understand. This is highlighted in red in the new submission.
For the selection of parameters: we selected the parameters based on the comparison of various parameters through the experimental results, the loss function applied Cross-entropy, Center loss, AM-Softmax and AAM-Softmax through and evaluate their performance, and found that AM-Softmax performs the best in the experiments; through the use of Adam and SGD optimizer to conduct experiments separately, and compare their performance in the experiments; through the use of Adam and SGD optimizer to conduct experiments separately, and compare their performance in the experiments. optimizers separately to compare their differences in terms of model convergence speed and performance.Adam has better performance relative to SGD and does not require manual tuning of the learning rate. Overall, these choices were made experimentally, based on model performance and stability during training. These choices may be influenced by factors such as the characteristics of the task, the characteristics of the dataset, and hardware resources, so the best choice may vary from case to case. Changes were made in lines 384-390 of the revised version.
Once again, we thank you for your valuable suggestions, and we are dedicated to improving the quality of the paper to meet your requirements.
Sincerely,
Xuebing Jing
Reviewer 3 Report (New Reviewer)
Comments and Suggestions for Authors
The manuscript provides some fusion models pre-trained on different datasets to verify people based on voice and facial embeddings. The manuscript describes the proposed approach clearly; the different sections of a research project are presented, and the contributions according to their testing and validation are promising.
However, a minor change is required regarding the abstract since it is not clear and does not mention the utility of the proposed models. What are the resolved issues?
Comments on the Quality of English LanguageMinor editing of English language required
Author Response
Dear Reviewer,
Thank you for your feedback. We appreciate your constructive comments. The abstract has been revised in the newly submitted manuscript, with red color has been marked. The main research question in this paper is to explore audio-visual multimodal feature fusion methods to improve personnel verification performance. The motivation for this research stems from the realization that unimodal verification systems face performance bottlenecks, especially in complex scenarios. This study examines how information from different modalities can be effectively integrated to improve the accuracy and robustness of a personnel verification system.
Thank you for your time and consideration.
Sincerely,
Xuebing Jing
Reviewer 4 Report (New Reviewer)
Comments and Suggestions for Authors
- They must add a comparative table of the methods or techniques that exist in the introduction section.
- In the results section they do not present the application of the method used, it would be important to apply it to something real.
- Under what conditions were the data used for the experiment acquired, what were they? How does it affect the application of the proposed method?
- They must review their English, there are words to improve in the document.
Comments on the Quality of English Language- They must review their English, there are words to improve in the document.
Author Response
Dear Reviewer,
Thank you for your valuable feedback. We have carefully considered your suggestions and made the following revisions to address your concerns:
Comparative Table in the Introduction Section: We have included in the introduction a comparative table of existing methods and techniques in Table 1. The table provides a quick overview of the different methods. The reader can now compare at a glance the main aspects of the various methods in lines 47-48.
Application of the Method in the Results Section: In the results section, we detail the real-world application of the proposed methodology. This addition provides the reader with a practical context that demonstrates how our methodology can be useful in a practically relevant setting. We believe that this improvement adds significant value to the overall presentation of our work in lines 461-472.
Clarification on Data Acquisition Conditions: The datasets for this experiment are all public datasets and are described in line 357-360 of the modified version.
Language Review: We have thoroughly reviewed the entire document and addressed language-related issues. We have refined the wording for clarity and coherence to ensure a smoother reading experience. If there are specific words or phrases you have in mind, please let us know, and we will make the necessary adjustments. Changes in the modified version are highlighted in red.
We appreciate your thorough review, and we believe these revisions enhance the quality and comprehensibility of our manuscript. If you have any further suggestions or specific language improvements, please feel free to share them.
Thank you for your time and consideration.
Best regards,
Xuebing Jing
Round 2
Reviewer 2 Report (New Reviewer)
Comments and Suggestions for Authors
Thank you very much for the changes you have made. In my opinion the work is now ready for publication.
This manuscript is a resubmission of an earlier submission. The following is a list of the peer review reports and author responses from that submission.
Round 1
Reviewer 1 Report
Comments and Suggestions for Authors
Multimodal biometric using speech and face recognition using attention based deep learning. The paper provided recent research of multimodal biometric fusion based method and compared their approach. Their EER rate to identify a subject is promising to produce a good impact on biometric authentication. Table 5 provided a comparison of their results with relevant research. They should also consider the processing complexity of the speech and face recognition, also Generative AI related hacking issues in biometric applications. Their arguments and experiment result provide good evidence and use case opportunity of fusion based biometric authentication. References are recent and relevant to the topic and usages. What is the limitation of their model, can they provide a section of limitation ?
Author Response
Dear reviewer,
Thank you for your decision and the constructive feedback on my manuscript. We have carefully considered your suggestions and made several revisions accordingly. We have made every effort to improve and modify the original manuscript. I have incorporated the changes in the latest submission of the manuscript based on your feedback.
We believe that these modifications have significantly enhanced the quality and clarity of the manuscript. We are grateful for the opportunity to address your comments and hope that the revised version meets your expectations.Thank you once again for your valuable input and consideration.
Sincerely,
Xuebing Jing
Reviewer 2 Report
Comments and Suggestions for Authors
1. The introduction appears more like a short review of the state of the art than an actual introduction. It should focus on justifying why a solution to this problem should be sought and why it is important to address it.
2. In the related works, a sample of some developed projects is presented, but there is no emphasis on what this implementation contributes or how this solution differs from others. This search should be enriched with recent works.
3. The article details the proposed model, but this model appears to be disconnected from the literature review, and the contribution of this solution compared to others is not demonstrated.
4. The implementation of the model is described, but the contributions compared to other models previously developed and described in the literature are not addressed."
5. Unfortunately, I do not find in the conclusion important aspects such as:
Recapitulation of results: Briefly summarize the main findings and results of the study.
Response to research objectives and questions: Demonstrating how the study's results address the research objectives and questions posed at the beginning of the article.
Interpretation of results: Analyzing and discussing the obtained results in the context of existing scientific literature, emphasizing similarities or differences found.
Implications and relevance: Highlighting the importance and relevance of the results for the field of study and their potential impact on the scientific community or society.
Limitations: Recognizing the study's limitations and mentioning aspects that could have affected the results.
Recommendations and future perspectives: Providing recommendations for future research and potential lines of study that arise from the current findings.
Conclusive closure: Providing a solid and conclusive closing that reinforces the key points and overall importance of the study.
Comments on the Quality of English LanguagePlease review the overall writing of the paper
Author Response
Dear reviewer,
Thank you for your decision and the constructive feedback on my manuscript. We have carefully considered your suggestions and made several revisions accordingly. We have put forth our utmost effort to improve and modify the original manuscript. I have addressed your comments in the response provided in the following link.
We believe that these modifications have significantly enhanced the quality and clarity of the manuscript. We are grateful for the opportunity to address your comments and hope that the revised version meets your expectations.Thank you once again for your valuable input and consideration.
Sincerely,
Xuebing Jing

Reviewer 3 Report
Comments and Suggestions for Authors
This paper brings a novelty to the scientists and researchers in the studied field. The ideas are clearly formulated, and the paper is well structured. As far as the research content is concerned, it is interesting and suitable for the readers of this journal. As an expert in the field, I consider the topic of the paper interesting. It fulfills all necessary quality standards. The tables are of a good quality, too. They are readable and the presented results are clear. Figures meet the formal requirements; they do not need to be changed. Authors of the paper provide the readers with plenty of relevant referenced papers. Based on all mentioned characteristics of the paper I recommend accepting and publish it in its current form.
Author Response
Dear reviewer,
Thank you for taking the time out of your busy schedule to write and express your appreciation for this paper. I am grateful for the acknowledgment from you, esteemed expert, and I am delighted to receive your correspondence. I wish you good health and success in your work.
Sincerely,
Xuebin Jing
Reviewer 4 Report
Comments and Suggestions for Authors
The paper proposes methods for audio-visual fusion based on attention mechanisms, including experimental results. The topic fits with the journal. Overall, the authors managed to convey the message behind their work with the current English usage - although it requires a careful proofreading. One aspect that I appreciated is the intuition behind the use of attention mechanisms for their goal. While the paper has strengths in terms of experiments, there are several critical issues that need to be addressed, especially to enhance its clarity, significance, and technical contribution.
Major Points:
Section 1: From a technical perspective, the paper lacks originality and significance, as it relies heavily on existing pre-trained models, well-known attention layers, and datasets. The authors need to highlight the unique aspects of their approach that distinguish it from existing methods. Specifically:
- The paper does not clearly articulate the open issues in prior work and how the proposed contributions uniquely fill these gaps. It is essential to provide a comprehensive understanding of the challenges faced in the field and how the proposed approach addresses them differently from prior work. Please clarify.
- The list of advantages for the proposed fusion method should be seamlessly integrated throughout the text rather than presented in a separate numbered list. This would allow readers to better appreciate the significance of the approach as they progress through the paper. Please adjust.
- The novel contributions of the paper are not clearly specified. It is crucial to explicitly state what novel elements or innovations the proposed method introduces, both technically and experimentally. Please add a bullet list with two/three points on this.
Section 2: The related work section, particularly in Section 2.2, appears to miss important knowledge regarding fusion solutions in the literature.
- Given the extensive nature of the topic, the authors should expand Section 2.2, taking into account recent surveys (e.g., https://www.sciencedirect.com/science/article/pii/S1566253522002081 and https://www.sciencedirect.com/science/article/pii/S156625351830839X) and methods (e.g., https://link.springer.com/chapter/10.1007/978-3-030-40014-9_7, https://www.scitepress.org/Link.aspx?doi=10.5220/0007690902550265, https://ieeexplore.ieee.org/document/9856650, and https://www.mdpi.com/1424-8220/23/13/5890) that provide a more comprehensive overview of audio-visual fusion techniques. Citing recent surveys and methods, including discussions on these developments, can enhance the paper's contextualization.
- Section 2.3, which discusses loss functions, appears to be a peripheral component compared to the main focus of the paper and can be removed to streamline the content.
Section 3:
- It is not clear why the MTCNN (Multi-task Cascaded Convolutional Networks) is relevant to the paper's focus and deserves a subsection per sé. The authors should clarify the significance of incorporating MTCNN within the text and not just citing it, to streamline the content. I suggest removing this paragraph.
Section 4:
- In Section 4.1.1, the description of data pre-processing should be presented in a more conceptual manner rather than resembling coding instructions. Providing a clear conceptual understanding of data pre-processing can benefit a wider range of readers.
- The use of only one face encoder and one speaker encoder should be justified or discussed in more detail. Please explain why this choice was made and whether it has any implications for the results. Moreover, the authors should include experiments with different face encoders and speaker encoders to show the transferability of the proposed fusion approaches.
- The choice of VoxCeleb1 over VoxCeleb2 as a dataset should be explained. Please justify why VoxCeleb1 was chosen as the dataset for experimentation. Furthermore, the authors should extend their experiments to cover also VoxCeleb2, which is becoming the consolidated state of the art for this task.
- The paper lacks comparison with other fusion baselines, aside from those proposed by the authors. The authors should include comparisons with existing fusion methods in the literature, such as those mentioned in the related work, to provide a stronger context for evaluating the proposed approach.
- The discussion of results in Section 4.2.2 appears chaotic and too long. The authors should, therefore, better organize it for clarity and coherence, shortening and aiming to emphasize the main observations only.
Additional Comments:
- Please consider improving reproducibility by sharing source code or detailed implementation guidelines to allow others to replicate the experiments.
- Abbreviations like "SV" and "FV" should be expanded upon when introduced to aid reader comprehension.
- Ensure that spaces are added between citations and the text for proper formatting (e.g., "ResNet34 [1]" and "loss (MMCosine)").
- To highlight the best results within tables, consider using bold formatting.
Comments on the Quality of English LanguagePlease see above.
Author Response

(The authors gave the same response as above.)

Round 2
Reviewer 4 Report
Comments and Suggestions for Authors
Thanks to the authors for their responses. Unfortunately the revised manuscript does not adequately address the main concerns included in my original review, including the lack of re-run all the unimodal fusion experiments, the lack of use of VoxCeleb 2 or other additional datasets, and no link to the source code, with justifications that have a limited scientific relevance.
Comments on the Quality of English LanguageSee above.